# Antibiotic collateral sensitivity is contingent on the repeatability of evolution

Daniel Nichol[1,2,12], Joseph Rutter[3], Christopher Bryant[4], Andrea M. Hujer[3,4], Sai Lek[5], Mark D. Adams[5], Peter Jeavons[1], Alexander R.A. Anderson[2], Robert A. Bonomo[3,4,6,7,8,9] & Jacob G. Scott [7,10,11]

Antibiotic resistance represents a growing health crisis that necessitates the immediate discovery of novel treatment strategies. One such strategy is the identification of collateral sensitivities, wherein evolution under a first drug induces susceptibility to a second. Here, we report that sequential drug regimens derived from in vitro evolution experiments may have overstated therapeutic benefit, predicting a collaterally sensitive response where cross-resistance ultimately occurs. We quantify the likelihood of this phenomenon by use of a mathematical model parametrised with combinatorially complete fitness landscapes for *Escherichia coli*. Through experimental evolution we then verify that a second drug can indeed stochastically exhibit either increased susceptibility or increased resistance when following a first. Genetic divergence is confirmed as the driver of this differential response through targeted and whole genome sequencing. Taken together, these results highlight that the success of evolutionarily-informed therapies is predicated on a rigorous probabilistic understanding of the contingencies that arise during the evolution of drug resistance.

[1] Department of Computer Science, University of Oxford, Oxford OX1 3QD, UK. [2] Department of Integrated Mathematical Oncology, H. Lee Moffitt Cancer Center and Research Institute, Tampa, FL 33612, USA. [3] Research Service, Louis Stokes Department of Veterans Affairs Hospital, Cleveland, OH 44106, USA. [4] Department of Medicine, Case Western Reserve University School of Medicine, Cleveland, OH 44106, USA. [5] The Jackson Laboratory for Genomic Medicine, 10 Discovery Dr, Farmington, CT 06032, USA. [6] Departments of Biochemistry, Molecular Biology and Microbiology, and Pharmacology, Case Western Reserve University School of Medicine, Cleveland, OH 44106, USA. [7] Center for Proteomics and Bioinformatics, Case Western Reserve University School of Medicine, Cleveland, OH 44106, USA. [8] Medicine Service and Geriatric Research Education and Clinical Center (GRECC), Louis Stokes Cleveland Department of Veterans Affairs Medical Center, Cleveland, OH 44106, USA. [9] CARES, CWRU-VA Center for Antibiotic Resistance and Epidemiology, Cleveland, OH 44106, USA. [10] Wolfson Centre for Mathematical Biology, Mathematical Institute, University of Oxford, Oxford OX1 3LB, UK. [11] Departments of Translational Hematology and Oncology Research and Radiation Oncology, Cleveland Clinic, Cleveland, OH 44195, USA. [12]Present address: Evolutionary Genomics and Modelling Lab, Centre for Evolution and Cancer, The Institute of Cancer Research, London, UK. Correspondence and requests for materials should be addressed to D.N. (email: daniel.nichol@icr.ac.uk) or to J.G.S. (email: scottj10@ccf.org)

The emergence of drug resistance is governed by Darwinian dynamics, wherein resistant mutants arise stochastically in a population and expand under the selective pressure of therapy[1]. These evolutionary principles underpin resistance to the presently most effective therapies for bacterial infections[2], cancers[3], viral infections[4] and disparate problems such as the management of invasive species and agricultural pests[5]. Biological mechanisms of drug resistance often carry a fitness cost in the absence of the drug and further, different resistance mechanisms can interact with one another to produce non-additive fitness effects, a phenomenon known as epistasis[6]. These trade-offs can induce rugged fitness landscapes, potentially restricting the number of accessible evolutionary trajectories to high fitness[7,8] or rendering evolution irreversible[9].

Identifying evolutionary trade-offs forms the basis of an emerging strategy for combating drug resistance; prescribing sequences of drugs wherein the evolution of resistance to the first induces susceptibility to the next[10–13]. Where this occurs, the first drug is said to induce collateral sensitivity in the second. Conversely, where the first drug induces increased resistance in the second, collateral (or cross) resistance has occurred. Recently, in vitro evolution experiments have been performed, in both bacteria[10,14–18] and cancers[19,20], to identify drug pairs or sequences exhibiting collateral sensitivity. One common protocol for these experiments proceeds by culturing a population in increasing concentrations of a drug to induce resistance, and then assaying the susceptibility of the resultant population to a panel of potential second-line therapies. From these experiments, sequences or cycles of drugs in which each induces collateral sensitivity in the next have been suggested as potential therapeutic strategies to extend the therapeutic efficacy of a limited pool of drugs[10,20]. For some cancer therapies, which often have severe side-effects and high toxicity, such sequential therapies may be the only way to combine the use of multiple drugs.

Drug pairs that are identified as collaterally sensitive in a small number of in vitro evolutionary replicates may not in fact induce collateral sensitivity each time they are applied. This hypothesis arises from the observation that evolution is not necessarily repeatable; resistance to a drug can arise through multiple different mechanisms, as has been observed in cancers[21] and bacteria[22]. Further, one mechanism may confer resistance to a second drug, whilst another may induce increased susceptibility, as was recently demonstrated in a drug screen of over 3000 strains of *Staphylococcus aureus*[23]. In previous experimental evolution studies to identify collateral sensitivity this phenomenon has been directly observed. For example, Barbosa et al.[24] observed contrasting collateral response in evolutionary replicates of *Pseudomonas aeruginosa*. Oz et al.[25] observed the same phenomenon in *E. coli* wherein a pair of evolutionary replicates was performed under exposure to the ribosomal (30S) inhibitor tobramycin, resulting in one exhibiting increased sensitivity to chloramphenicol and one exhibiting increased resistance. Similar effects are evident in cancer studies. Zhao et al.[19] observed that the sensitivity of a BCR-ABL leukaemia cell line to cabozantinib can both increase and decrease following exposure to bosutinib, and identified a single-nucleotide variation responsible for this differential collateral response.

The extent of the impact of differential collateral response on the design of sequential drug therapies is not yet fully understood. Here, we provide a clear evolutionary explanation for differential patterns of collateral repsonse through a combination of mathematical modelling and experimental evolution. Through mathematical modelling we demonstrate the extent to which the existence of multiple evolutionary trajectories to drug resistance can render collateral sensitivities stochastic, and discuss the implications for in vitro experimental evolution. We next

empirically demonstrate the existence of multiple trajectories in the evolution of *E. coli* through in vitro experimental evolution. Previous studies have explored the collateral repsonse by considering all pairs from a pool of antibiotics, each with a small number of evolutionary replicates[10,14,15,17]. We instead perform 60 parallel evolutionary replicates of *E. coli* under cefotaxime to demonstrate the extent of heterogeneity in second-line drug sensitivity. Through genomic sequencing we confirm that different mutations (i.e., different evolutionary trajectories) are responsible for this heterogeneity. Critically, we find that collateral sensitivity is never universal, and is in fact rare. Finally, we derive collateral sensitivity likelihoods which we argue are critical statistical benchmarks for the clinical translation of sequential drug therapies.

## Results

**Mathematical modelling of evolution.** The potential impact of divergent evolution can be conceptualised in the classical fitness landscape model of Wright[26], wherein genotypes are projected onto the two dimensional $x$–$y$ plane and fitness represented as the height above this plane. Evolution can be viewed as a stochastic 'up–hill' walk in this landscape wherein divergence can occur at a saddle. Figure 1 shows such a schematic fitness landscape annotated to demonstrate the capacity for divergent evolution and the potential effects on collateral sensitivity.

Previous studies have attempted to empirically determine the structure of the fitness landscape for a number of organisms and under different drugs[27]. In these studies, a small number of mutations associated with resistance are first identified. Strains are engineered corresponding to all possible combinations of presence and absence of these mutations and the fitness of each strain is measured by a proxy value, for example minimum inhibitory concentration (MIC) of a drug or average growth rate under a specific dose. These measurements are combined with the known genotypes to form a fitness landscape. However, to derive fitness landscapes through this method, the number of strains that must be engineered grows exponentially with the number of mutations of interest. Thus only small, combinatorially complete, portions of the true fitness landscape can be measured, for example, consisting of 2–5 alleles[7,27,28]. Nevertheless, these restricted fitness landscapes can provide valuable insight into the evolution of drug resistance.

Mira et al.[29] derived fitness landscapes for *E. coli* with all combinations of four fitness conferring mutations (M69L, E104K, G238S and N276D) in the TEM gene and measured fitness under 15 different $\beta$-lactam antibiotics (See Supplementary Fig. 1, Supplementary Table 1), using the average growth rate (over 12 replicates) as a proxy of fitness. Of these 15 landscapes, 14 were identified as having multiple local optima of fitness, indicating the potential for the divergence of evolutionary trajectories. We utilised these landscapes, coupled with mathematical modelling[12] (see Methods), to estimate the likelihood of the different evolutionary trajectories from a wild-type genotype (denoted 0000) to each of the fitness optima. Using this model, we performed in silico assays for collateral sensitivity, mirroring the approach taken by Imamovic and Sommer[10] (Fig. 2). For each drug, we first stochastically simulated an evolutionary trajectory from the wild-type genotype to a local fitness optimum genotype and then, for all other landscapes, compared the fitness of this local optimum genotype to that of the wild-type. A schematic of this simulation is shown in Fig. 2a. Figure 2b shows an example of two evolutionary trajectories that can arise stochastically in this model under the fitness landscape for ampicillin.

We exhaustively enumerated all tables of collateral response that can arise under this model (see Supplementary Figs. 2–10 for

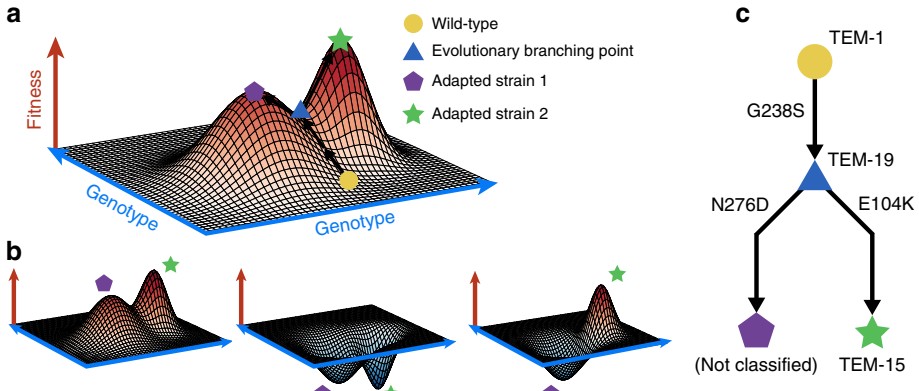

**Fig. 1** Evolutionary saddle points can drive divergent collateral response. **a** A schematic fitness landscape model in which divergent evolution can occur. Following Wright[26], the x–y plane represents the genotypes and the height of the landscape above this plane represents fitness. Two evolutionary trajectories, both starting from a wild-type genotype (yellow circle), are shown. These trajectories diverge at an evolutionary saddle point (blue triangle) and terminate at distinct local optima of fitness (purple pentagon, green star). As the saddle point exists, evolutionary trajectories need not be repeatable. **b** Schematic landscapes for a potential follow-up drug are shown, the collateral response can be (from left to right): always cross-resistant, always collaterally sensitive, or dependent on the evolutionary trajectory that occurs stochastically under the first drug. **c** A potential evolutionary branching point in the TEM gene of *E. coli* identified in the fitness landscape for cefotaxime derived by Mira et al.[29]

further details). Figure 2c shows the best case (most susceptible following evolution), worst case (highest resistance following evolution) and mostly likely collateral response tables that arose in this analysis, along with the mean collateral response table (expectation of collateral response for each pair). This analysis suggests that there is remarkable variation in collateral response arising solely from the stochastic nature of mutation that ultimately drives evolution under a first drug. Indeed, we find a total of 82,944 unique tables can arise, of which the most likely occurs with probability 0.0023. Amongst the 225 ordered drug pairs, only 28 show a guaranteed pattern of collateral sensitivity, whilst a further 94 show a pattern of guaranteed cross-resistance. For 88 pairs, the first drug can induce either collateral sensitivity or cross-resistance in the second as a result of divergent evolution under the first drug. Critically, if a collateral response table is generated by stochastic in silico simulation, and a collaterally sensitive drug pair chosen at random from this table, then the expected probability that first of these two drugs will induce cross-resistance in the second is 0.513 (determined from $10^6$ simulations of this process).

**Experimental evolution induces heterogeneous collateral response.** The mathematical model used above represents a simplification of biological reality as the assumption of a monomorphic population need not hold and the parametrisation is made using incomplete fitness landscapes. To experimentally validate our predictions, we verified the existence of divergent collateral response through experimental evolution. Mirroring previous experimental approaches[10,16,18–20], we performed in vitro evolution of *E. coli* (strain DH10B carrying phagemid pBC SK(−) 198, expressing the *beta*-lactamase gene SHV-1) in the presence of the β-lactam antibiotic cefotaxime. Bacterial populations were grown using the gradient plate method with concentrations of cefotaxime varying between approximately 0.1 and 1000 μg ml$^{-1}$ and 256 μg ml$^{-1}$ over a course of 10 passages lasting 24 h (see Fig. 3a and Methods for details). In total, 60 replicates of experimental evolution were performed. We denote the resulting populations by X1–X60. For replicates X1–X12, aliquots were taken following each second passage and the MIC to a panel of second-line drugs assayed. A time-series for the MIC of X1–X12 replicates under cefotaxime is shown in Fig. 3b.

As expected, the replicates exhibit increased resistance to cefotaxime over the ten passages, although with varying magnitude and different trajectories.

For each of a panel of eight second-line antibiotics (Table 1), the MIC for the replicates X1–X60 was determined following passage ten, in addition to the MIC for the parental strain (Supplementary Dataset 1, Methods). Figure 4 shows how the MICs of X1–X60 differ from the parental line. As predicted, we find that the collateral change in sensitivity is highly heterogeneous, and show that both collateral sensitivity and cross-resistance can arise to the antibiotics piperacillin (PIP), ticarcillin/clavulanate (TCC) and ampicillin/sulbactam (AMS).

**Genomic profiling reveals divergent evolution.** Differential patterns of drug resistance could be driven by the different replicates having experienced different numbers of sequential mutations along a single trajectory wherein each induces a shift in response (temporal collateral sensitivity[19]), by evolutionary divergence at a branching point in the landscape or by non-genetic mechanisms of resistance. To elucidate the underlying mechanisms, we first performed targeted sequencing of the SHV gene for each of the 10 passage time points for 12 evolutionary replicates (X1–X12) (Fig. 3b). Through this analysis we identified five variants of SHV-1 amongst the 12 replicates. X1, X5, X7–X9 and X11 all possess wild-type SHV-1, X2 possesses the substitution G242S, X3 possesses G238C, X4 and X6 both possess G238A, and X10 and X12 both possess G238S. This analysis revealed no evidence of double substitutions in SHV, indicating a minimum of four fitness conferring substitutions that can occur in SHV-1 during exposure to cefotaxime, and confirming the existence of a multi-dimensional evolutionary branching point in the fitness landscape. Further, the sensitivity of the population to a second drug appears to be (at least partially) dependent on which of these substitutions occurs (Fig. 3, 4). For example, replicate X3 (harbouring G238C) exhibits a significant increase in susceptibility to TIC, PIP and SAM, whilst those replicates found to harbour wild-type SHV-1, or the other SNVs, exhibit either cross-resistance or no significant change in susceptibility to these drugs.

Through targeted sequencing of SHV alone we cannot not exclude the possibility that mutations in other genes, or large scale genomic alterations such as insertions or deletions, drive

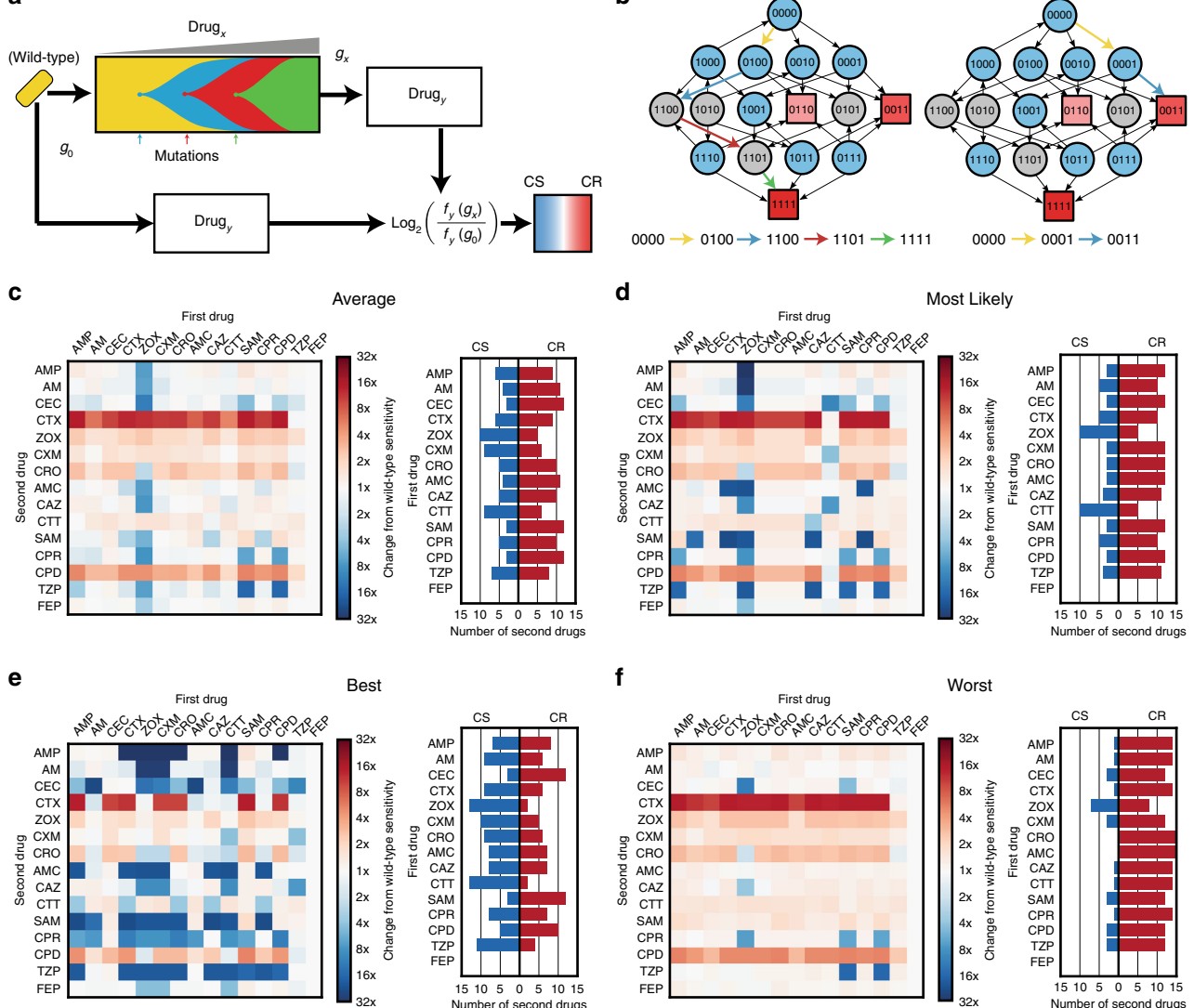

**Fig. 2** Mathematical modelling predicts highly variable collateral response. **a** A schematic of the model used to derive collateral response. Sequential mutations are simulated to fix in the population until a local optimum genotype arises. The fitness of this resultant genotype is compared to the fitness of the wild-type genotype for each of the panel of antibiotics. **b** The landscape for ampicillin derived by Mira et al.[29] represented as a graph of genotypes. Arrows indicate fitness conferring mutations between genotypes represented as nodes. Blue nodes indicate genotypes from which evolution can stochastically diverge, grey nodes indicate genotypes from which there is only a single fitness conferring mutation. Squares indicate local optima of fitness with colour indicating the ordering of fitness amongst these optima (darker red indicates higher fitness). Two divergent evolutionary trajectories, in the sense of the model shown schematically in (**a**), are highlighted by coloured arrows. **c–f** The average, most likely, best case, and worst case tables of collateral response derived through stochastic simulation. Columns indicate the drug landscape under which the simulation was performed and rows indicate the follow-up drug under which the fold-change from wild-type susceptibility is calculated. Bar charts indicate, for each labelled first drug, the number of follow-up drugs exhibiting collateral sensitivity (blue) or cross-resistance (red) in each case. CS - collaterally sensitive, CR - cross resistant

further divergence in collateral response. To explore whether additional background mutations arose during selection, we produced draft genome sequences for the replicates X1–X12 after passage 10 and looked for evidence of additional mutations. This genomic data confirmed the SHV-1 mutations found by sequencing of PCR products as described above. Nine of the twelve replicates contained additional mutations that include single-nucleotide variants (SNVs), large (>5 kb) deletions, and replicate-specific sites for insertion of IS1D (Table 2). *OmpC* encodes a membrane surface protein and *envZ* is responsible for osmoregulation by regulation of the expression of OmpC and other membrane proteins[30]. This suggests that drug resistance in X8, X9, X10 and X11 may be driven by mutations that result in restricted drug uptake at the cell membrane. Indeed, mutations

in *envZ* and cell surface proteins have been previously implicated as drivers of antibiotic resistance[31–33]. Stress-regulation through osmoregulation has been previously identified as inducing a trade-off with nutritional competence[34], suggesting that although these replicates do not exhibit collateral sensitivity, the resistant cells could face a fitness cost in the absence of drug. Similar patterns of fitness trade-off have been exploited in cancer treatments by using dose-modulation (adaptive) therapies that extend survival by inducing competition between sensitive and resistance cells[35–37]

We conclude that mutations in SHV-1 are the primary drivers of cefotaxime resistance as they are associated with the most substantial increases in MIC. For example, for replicate X12, which exhibits the highest endpoint MIC, no additional

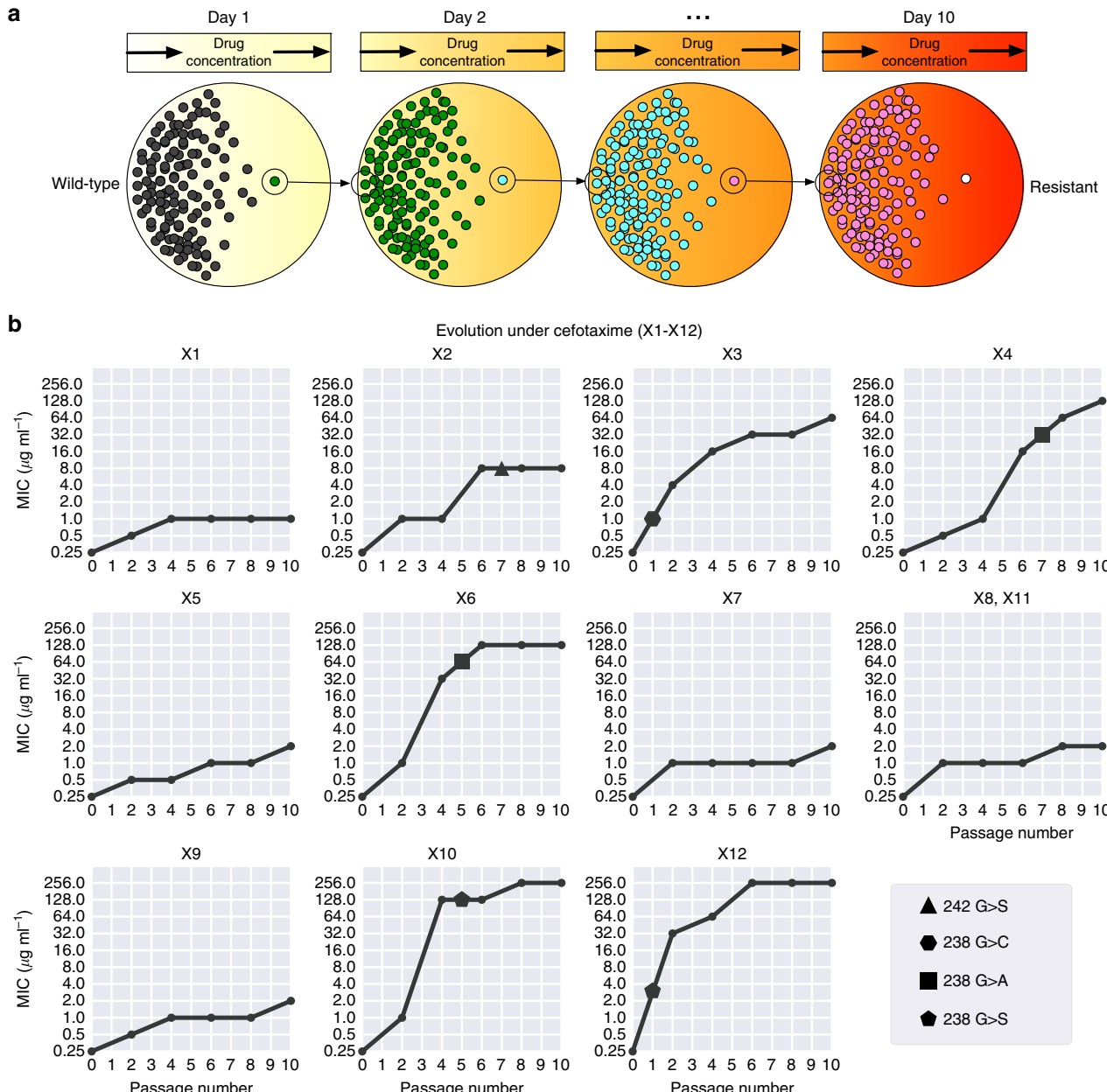

**Fig. 3** Experimental evolution reveals divergent collateral response. **a** A schematic of the evolutionary experiment. *E. coli* were grown using the gradient plate method and passaged every 24 h for a total of 10 passages. Sixty replicates of experimental evolution were performed. **b** The MIC for 12 replicates (X1–X12) under cefotaxime exposure was measured following passages 0, 2, 4, 6, 8 and 10. These values are plotted, revealing heterogeneity in the degree of resistance evolved to cefotaxime. Targeted sequencing of the SHV gene was performed following each passage revealing four different SNVs between the replicates marked by geometric shapes (triangle—G242S, hexagon—G238C, square—G238A and pentagon—G238S). Mutations are marked at the earliest time point they were detected in each replicate

| Table 1 Antibiotic drugs used in this study | | | |
|---|---|---|---|
| **Antibiotic** | **Abbreviation** | **Antibiotic group** | **Notes** |
| Cefotaxime | CTX | Cephalosporin | |
| Ciprofloxacin | CIP | Fluoroquinolone | |
| Ampicillin/sulbactam | SAM | $\beta$-lactam combination | 2:1 ratio of ampicillin to sulbactam |
| Gentamicin | GNT | Aminoglycoside | |
| Ticarcillin/clavulanate | TIC | $\beta$-lactam combination | 2 $\mu$g ml$^{-1}$ clavulanate |
| Phosphomycin | PMC | Phosphomycin | |
| Ceftolozane/tazobactam | CFT | $\beta$-lactam combination | 2:1 ratio of ceftolozane to tazobactam |
| Piperacillin | PIP | Penicillin | |
| Cefazolin | CFZ | Cephalosporin | |

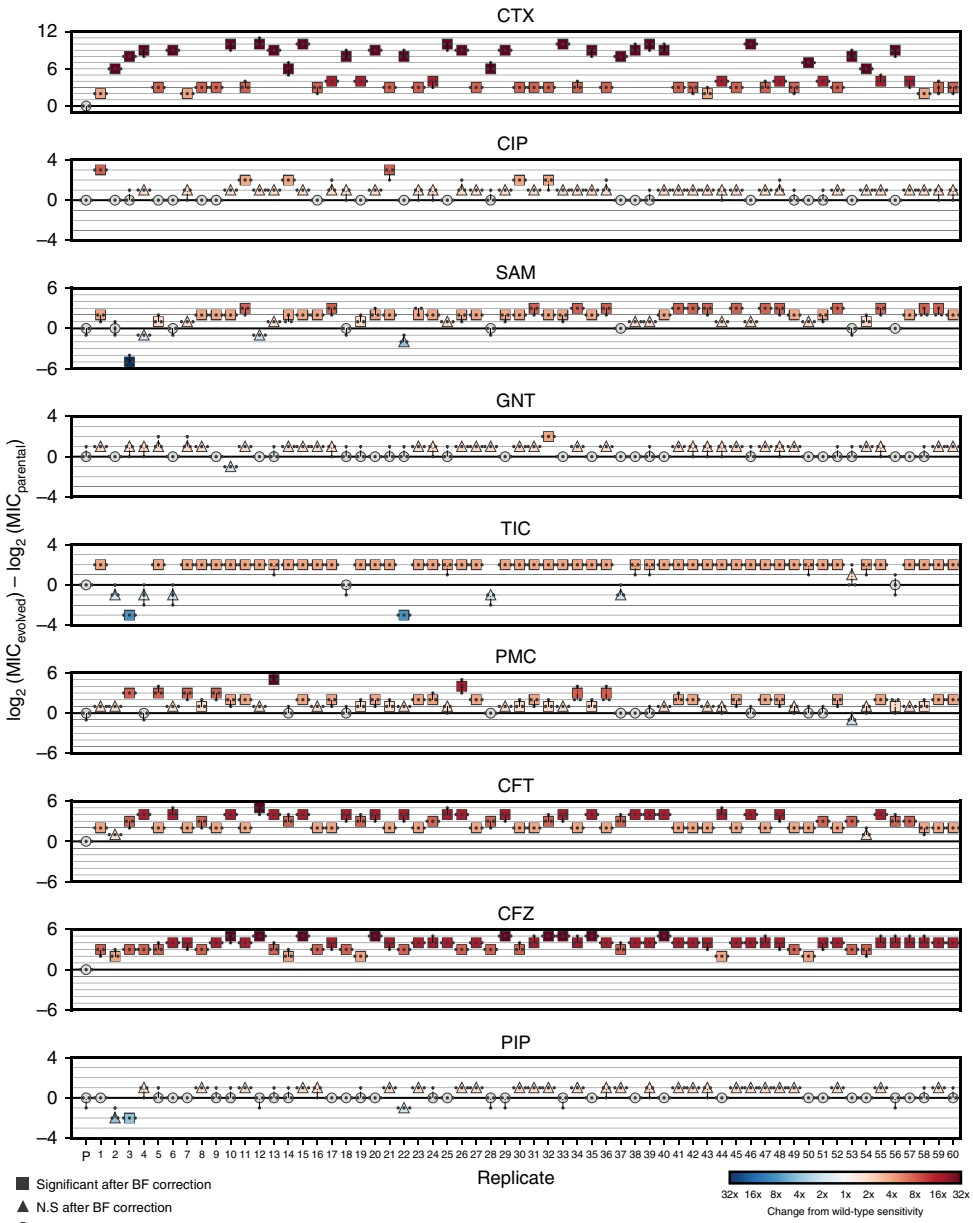

**Fig. 4** Collateral response following evolution under cefotaxime. The maximum likelihood estimates for the MICs of replicates X1–X60 under cefotaxime and eight other antibiotics. Small markers indicate individual measurements (taken in triplicate). Marker colour indicates fold-change from wild-type sensitivity (increased sensitivity—blues, increased resistance—reds). Significance is determined via a likelihood ratio test (see Methods) and Bonferroni (BF) corrected. Precise *p* values are reported in Supplementary Dataset 1

mutations were detected. In contrast, X1, X5, X8, X9 and X11 all had genomic mutations, lacked SHV-1 variants and had the lowest final cefotaxime MIC. We excluded the possibility of amplifications of SHV-1 by consideration of read depth ratios. The ratio of reads mapped to the gene and reads mapped to the plasmid backbone was very similar across all samples. The ratio of plasmid reads to chromosomal reads did differ across samples, but the fraction of plasmid-derived reads did not correlate with the MIC for cefotaxime (Supplementary Dataset 2) and is more likely due to variation in extraction efficiency for chromosomal versus plasmid DNA. We excluded the possibility of amplifications or deletions in chromosomal genes by consideration of read depth ratios (Supplementary Fig. 11).

We note that X7 exhibits an increase in resistance to cefotaxime without any associated genomic alterations. Similarly

X1, X5, X9 and X12 exhibit mutations, but none that are known to be associated with antibiotic resistance. Thus, we can infer that physiological adaptation or epigenetic adaptation may also be driving resistance to cefotaxime.

**Collateral sensitivity likelihoods.** Our experimental results demonstrate that the evolution of antibiotic resistance is non-repeatable, and that the efficacy of a second-line drug can depend on the specific evolutionary trajectory that occurs under a first. As such, where a pair of drugs exhibit collateral sensitivity in a small number of experimental replicates, it need not be the case that collateral sensitivity always occurs. Rather than give up entirely on the concept of collateral sensitivity between drugs, we propose that collateral sensitivity likelihoods (CSLs) should be derived[38]. By deriving the likelihood of collateral sensitivity between drugs,

| Table 2 Mutations identified through whole-genome sequencing | | | | |
|---|---|---|---|---|
| **Replicate** | **SHV-1 SNVs** | **Chromosomal SNVs** | **Deletions (ranges)** | **IS1D insertions** |
| **Parental** | | 2099555 T > C (intergenic yedK/yedL) | | |
| **X1p10** | | | 4166399–4177327 | |
| **X2p10** | G242S | | | |
| **X3p10** | G238C | | 3079240–3088253 | IS1D at 2849873 interrupts CP4-57 prophage predicted protein; 580 bp deletion adjacent |
| **X4p10** | G238A | | 3892703–3903946 2896300–2906979 | |
| **X5p10** | | | | IS1D at 3506340 interrupts dusB |
| **X6p10** | G238A | | | |
| **X7p10** | | | | |
| **X8p10** | | 2401329 T > A (ompC Q144V) | | |
| **X9p10** | | | | IS1D at 2401801 (upstream of ompC) |
| **X10p10** | G238S | 3630620 C > A (envZ R339L); 771931 C > T (speF L115L) | 4387943–4410705 | IS1D at 4410705 interrupts rpiB; 14 kb deletion adjacent |
| **X11p10** | | 3630620 C > A (envZ R339L) | 2896300–2906979 | IS1D at 2906979 interrupts gshA; 12 kb deletion adjacent |
| **X12p10** | G238S | | | |

The single-nucleotide variants (SNVs), insertions and deletions identified through whole-genome sequencing of the replicates X1–X12 following passage 10 are listed

we can quantify the risk associated with different drug sequences. Figure 5a shows an example table of collateral sensitivity likelihoods derived from the in silico evolution model. We note that whilst there exist 28 drug pairs exhibiting guaranteed collateral sensitivity ($P = 1.0$, right), there also 16 others with likelihood $1.0 > P > 0.75$ of collateral sensitivity. Where collateral sensitivity is assayed from a small number of experimental evolution replicates, these drug pairs may appear to exhibit universal collateral sensitivity and could thus unexpectedly fail stochastically. Conversely, if no universally collaterally sensitive drugs were known, drug pairs exhibiting a high likelihood of collateral sensitivity might represent the best option available.

Figure 5b shows the experimentally derived CSLs for antibiotics administered following cefotaxime. We find that collateral sensitivity is rare, with $P = \frac{1}{30}$ for TIC being the most likely. If we also consider the likelihood that sensitivity of the second-line drug is unchanged, then it is clear that PIP or gentamicin (GNT) are the best second-line drugs following cefotaxime (amongst those we have assayed). Conversely, cross-resistance is near universal in cefazolin and ceftolozane/tazobactam. For puromycin and ampicllin/sulbactam (SAM), we estimate that cross-resistance occurs with probability $P > 0.5$, but that the probability of no change or collateral sensitivity is still high ($P > 0.3$ in both cases). Drugs such as these highlight the importance of deriving collateral sensitivity likelihoods by means of multiple evolutionary replicates, as a single evolutionary replicate may identify unchanged sensitivity where cross-resistance is likely.

## Discussion

We have demonstrated the existence of an evolutionary branching point in the fitness landscape of *E. coli* under cefotaxime that can induce divergent evolution and differential collateral response to second-line antibiotics. By means of 60 replicates of experimental evolution, we have estimated the likelihood of collateral sensitivity in each of eight second-line therapies. Critically, we find that collateral sensitivity is never universal, and is in fact rare. Furthermore, by consideration of a mathematical model of evolution parametrised by small, combinatorially complete fitness landscapes, we have highlighted the extent and importance of evolutionary divergence. This modelling highlights that divergent collateral response is likely common (occurring in 14/15 drugs for which empirical landscapes were derived) and further, that even where collateral sensitivity is reported from a small number of evolutionary replicates, cross-resistance can still occur with high likelihood.

Taken together, our results indicate that we must take care when interpreting collateral sensitivity arising in low-throughput evolution experiments. To this end, we propose that collateral sensitivity likelihoods should be evaluated by use of multiple parallel evolutionary replicates to better capture the inherent stochasticity of evolution. The high-throughput experimental evolution necessary to accurately evaluate CSLs between many drug pairs could be facilitated by automated cell culture systems, such as the morbidostat developed by Toprak et al.[39], which incorporates automated optical density measurements and drug delivery to track and manipulate evolution.

It should be noted that although the evolution of pathological bacteria within the clinic is most likely stochastic, it is unclear whether the gradient plate system model used in the present study, and others[10], correctly captures this stochasticity. The gradient plate method proceeds by serial replating of bacterial populations that induces population bottlenecks and strong selection. This mode of population dynamics clearly differs from that which *E. coli* experience naturally. We note that our experimental results are derived only for the gradient plate method and that other protocols without serial passaging have also been explored[13]. Such experimental designs may exhibit less stochastic dynamics and thus permit the derivation of collateral sensitivity likelihoods with fewer replicates. Alternatively, it may be the case that additional stochasticity is introduced as evolutionary phenomena such as clonal interference, wherein multiple fitter clones compete, do not occur. To empirically determine collateral sensitivity likelihoods it may be the case that we must employ novel in vitro experimental techniques to more closely match in vivo dynamics. Here too, automated culture systems such as the morbidostat could help, as automated changes to the drug concentration can prevent the bacterial population expanding too rapidly, mitigating the need for serial replating and permitting high-throughput experiments.

The mathematical model we have presented does not capture the full complexity of evolution. For example, we do not account for deletions, insertions or duplications of genes such as SHV. Nevertheless, this model still proves useful in providing intuition about the extent to which stochasticity can drive differential collateral response. We can expect the introduction of additional mutational complexity to introduce further stochasticity. An immediate improvement to our modelling would be to extend the model to account for alternative population dynamics; for example, permitting heterogeneous populations, variable population sizes or drug pharmacodynamics. A further complication

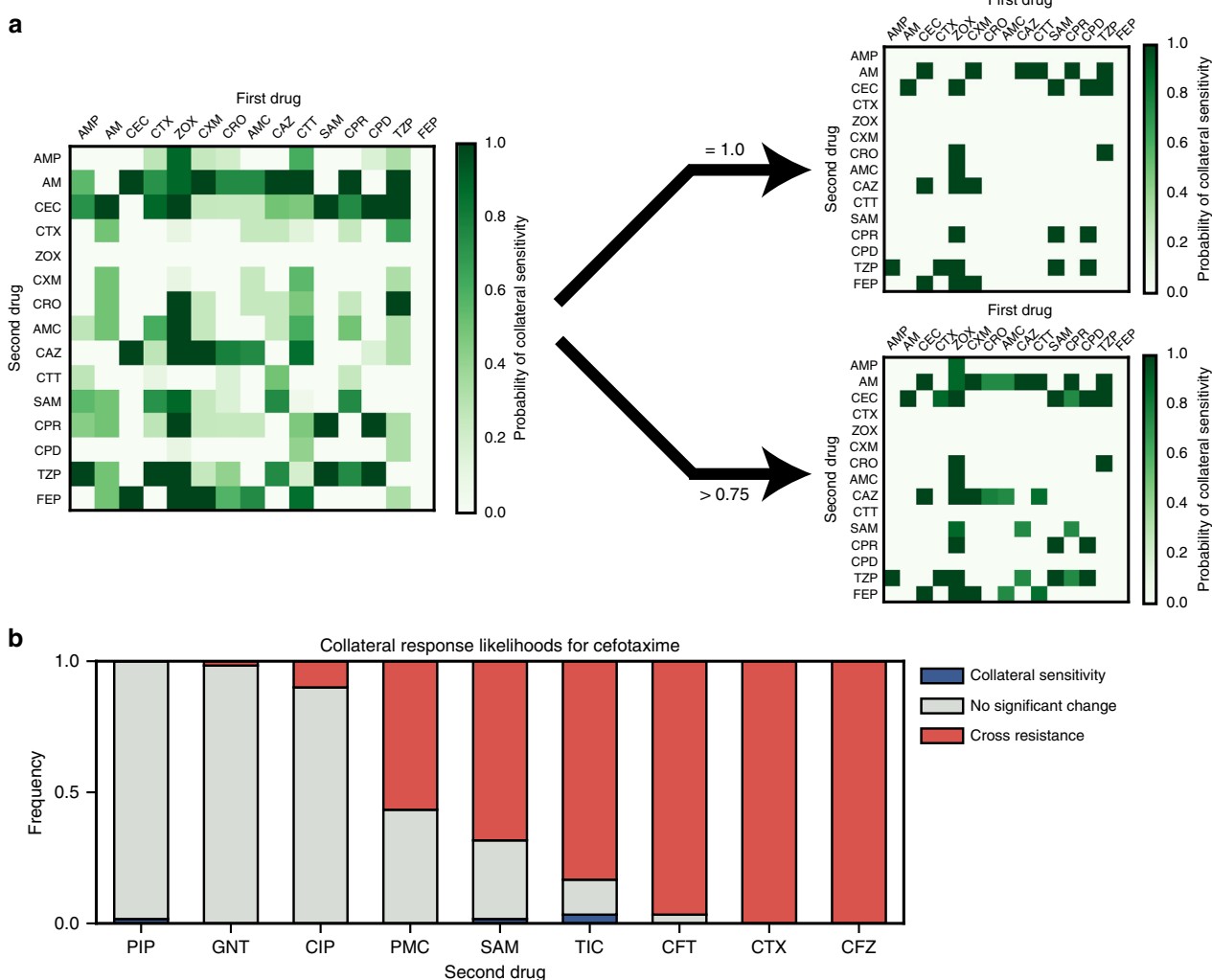

**Fig. 5** Collateral sensitivity likelihoods. **a** (Left) The table of collateral sensitivity likelihoods (CSLs) derived from the mathematical model. Each entry indicates the likelihood that the first drug (columns) induces increased sensitivity in the second (rows). (Right) The CSL table thresholded for drugs with $P = 1.0$ (top) and $P > 0.75$ (bottom) probability of inducing collateral sensitivity. **b** The estimated likelihoods for collateral sensitivity, cross-resistance or no change in sensitivity derived from the 60 replicates of experimental evolution

is that drug resistance can arise by physiological adaptions in addition to genetic mutation, which our mathematical modelling does not take into account. We see evidence for physiological adaption in the evolution of the replicate X7 which exhibits increased resistance to cefotaxime without associated mutations. Further, changes in sensitivity arising from such phenotypic plasticity may be reversible over short time scales[20]. Ultimately, by the use of extended mathematical models we may be able to better simulate in vitro experiments in order to understand how generalisable they are to in situ evolutionary dynamics[40].

As an alternative to high-throughput evolutionary experiments, we note that drug sequences are frequently prescribed in the clinic. Thus, the distributed collection of matched pre- and post-therapy drug sensitivity assays, potentially coupled with genomic sequencing where this is feasible, could provide sufficient data to determine CSLs. This approach is particularly appealing as the CSLs derived would not be subject to the caveats regarding experimentally derived measures of collateral sensitivities outlined above. Further, clinically derived CSLs would readily account for non-genetic adaptations and inter-patient variabilities in physiology that may impact drug sensitivities. A similar approach has already been employed in the treatment of HIV to monitor the evolution of drug resistance[41,42].

Regardless of the approach taken to derive CSLs, what is clear is that we must move beyond the present methodology of designing drug sequences through low-replicate-number experimental evolution, and towards an evolutionarily informed strategy that explicitly accounts for the inherent stochasticity of evolution.

## Methods

**Mathematical modelling of evolution**. The probabilities for evolutionary trajectories through the empirically derived fitness landscapes were calculated from a previously published mathematical model[12]. Briefly, the population is assumed to be isogenic and subject to strong selection weak mutation evolutionary dynamics. Thus, the population genotype (taken from domain $\{0, 1\}^4$) is modelled as periodically replaced by a fitter (as determined by the landscape) neighbouring genotype (defined as any genotype whose Hamming distance from the population genotype is equal to one). This process is stochastic and the likelihood of a genotype, $j$, replacing the present population genotype, $i$, is given by

$$\mathbb{P}(i \rightarrow j) = \begin{cases} \dfrac{(f(j) - f(i))^r}{\sum\limits_{\substack{g \in \{0,1\}^N, \mathrm{Ham}(i,g) = 1 \\ f(g) - f(i) > 0}} (f(g) - f(i))^r} & \text{if } f(j) > f(i) \text{ and } \mathrm{Ham}\,(i, j) = 1 \\ 0 & \text{otherwise.} \end{cases} \tag{1}$$

Where no such fitter neighbour exists, the process is terminated. The value of $r$ determines the extent to which the fitness benefit of a mutation biases the

likelihood that it becomes the next population genotype. We take $r = 0$, corresponding to fixation of the first arising resistance conferring mutation, but our results are robust to changes in $r$ (see Supplementary Note 1 for details).

For the simulations of in vitro evolutionary experiments, we assume an initial genotype of $g_0 = 0000$ and determine the final population genotype by sampling from the model until termination at a local optimum of fitness, say $g^*$. Simulated collateral response was calculated as the fold difference between $g_0$ and $g^*$ in a second fitness landscape. Collateral response outcomes for all drug pairs are shown in Supplementary Figs. 2–10.

**Experimental adaptation to cefotaxime**. All 60 evolutionary replicates were derived from *E. coli* DH10B carrying phagemid pBC SK(−) expressing the *β*-lactamase gene SHV-1[43]. The SHV-1 *β*-lactamase gene was subcloned into pBC SK(−) (Stratagene) from a clinical strain of *Klebsiella pneumoniae* 15571. In brief, a 1384 bp ScaI-ClaI DNA fragment containing the upstream flanking sequence, promoter, ribosomal binding site and intact open reading frame was cloned into pBC SK(−) at the EcoRV-ClaI sites. This clone was transformed into *E. coli* DH10B (ElectroMAX, Invitrogen).

Using a spiral plater, cefotaxime solutions were applied to Mueller Hinton (MH) agar plates in a continuous, logarithmic dilution to achieve a radial concentration gradient of antibiotic from approximately 0.1–1000 µg ml$^{-1}$. *E. coli* DH10B pBCSK(−) *bla*SHV-1 colonies were suspended to a concentration of 7log10 CFU ml$^{-1}$ in MH broth. Antibiotic plates were then swabbed along the antibiotic gradient with the bacterial suspension. Plates were incubated overnight at 37 °C. The most resistant colonies, as measured by the distance of growth along the gradient, were resuspended and used to swab a freshly prepared gradient plate. The process was repeated for a total of ten passages. The entire experiment was completed 60 times using the same parental strain to generate the cefotaxime resistance replicates X1–X60.

**Determination of minimum inhibitory concentration**. The minimum inhibitory concentration of each antibiotic was determined for both the parent strain and the cefotaxime resistant replicates according to guidelines outlined by the Clinical and 314 Laboratory Standards Institute[44]. Briefly, bacterial strains were grown 18–20 h in MH broth in a shaking incubator at 37 °C. Cultures were diluted and an inoculum replicator used was to deliver $10^4$ CFU to the surface of MH agar plates containing antibiotic. Plates were incubated at 37 °C for 16–20 h. The MIC was taken as the lowest concentration of antibiotic that completely inhibited growth. MICs were assayed in triplicate as series of twofold dilutions. Where the MIC exceeded the maximum concentration considered, 4096 µg ml$^{-1}$, the precise value was not determined and a lower bound MIC of ≥8192 µg ml$^{-1}$ was taken.

The MIC was determined from the replicates by maximum likelihood estimation using a statistical model outlined by Weinreich et al.[7]. Briefly, we assume that the $j$th $\log_2$ transformed MIC measurement for the $i$th evolutionary replicate, under the drug $d$, denoted $x_{i,j}^d$, is determined as

$$x_{i,j}^d = m_i^d + \epsilon_{i,j,d}, \tag{2}$$

where $\epsilon_{i,j,d} = +1, 0, -1$ with probability $e/2, 1 - e, e/2$, respectively. Here, each $m_i^d$ denotes the true MIC for the $i$th replicate (with $i = 0$ denoting the parental line) and $e$ denotes the likelihood of measurement error. We assume $e$ is fixed across technical replicates, evolutionary replicates and drugs. Note the assumption that we never erroneously take a measurement that differs from the true MIC by greater than a factor of two. This is justified by noting that in no instance do the maximum and minimum MICs measured in our analysis differ by greater than 4× (see Supplementary Dataset 1).

Maximum likelihood estimates (mle) for $m_i^d$ are used as the MICs in our analysis. The likelihood function is given by

$$\mathcal{L}\left(x_{0,1}^1 \dots x_{60,3}^9 \mid m_1^1 \dots m_{60}^9, e\right) = \prod_{d=1}^{9} \prod_{i=0}^{60} \prod_{j=1}^{3} \left( (1-e)\delta_{x_{i,j}^d, m_i^d} + \frac{e}{2}\delta_{x_{i,j}^d, m_i^d + 1} + \frac{e}{2}\delta_{x_{i,j}^d, m_i^d - 1} \right), \tag{3}$$

where $\delta$ denotes the Kronecker delta function. By observation, the mle for each $m_i^d$ is given by the median of $x_{i,1}^d, x_{i,2}^d$ and $x_{i,3}^d$, except in the case that two of these values are precisely 4× or 1/4× the other, in which case the mle is the mid-point between the maximum and minimum. Letting $r$ denote the number of replicate/drug combinations in which all three measurements equal the mle, $s$ denote the number in which 2/3 measurements equal the mle, $t$ the number in which 1/3 equal the mle and $u$ the number in which 0/3 equal the mle. Then the mle for $e$ is given by

$$e = \frac{s + 2t + 3u}{3(r + s + t + u)}. \tag{4}$$

This identity can be verified by first principles (by taking the derivative of the likelihood function) but is also quite intuitive—it is simply the proportion of measurements that differ from the inferred mle for the MIC. In our experiment, $r = 338$, $s = 196$, $t = 11$ and $u = 4$, which yields an mle for the measurement error rate of $e = 0.14$.

**Collateral sensitivity analysis and significance testing**. To determine collateral sensitivity (or cross-resistance) we determined which evolutionary replicates exhibited a significantly different MIC from the parental line via a likelihood ratio test. In total, 60 comparisons were performed for each of the 9 drugs, yielding a total of 540 comparisons. A Bonferroni correction was used to account for multiple hypothesis testing. For those replicates exhibiting a significant ($p < 0.05/540$) change in MIC, the collateral response was determined as

$$CR = m_i^d - m_0^d. \tag{5}$$

Otherwise, we set $CR = 0$.

**Targeted sequencing of SHV**. Plasmid DNA was isolated using the Wizard Plus Minipreps DNA purification systems (Promega). Sequencing of the SHV gene was performed using M13 primers (MCLab, Harbor Way, CA).

**Whole-genome sequencing**. For genome sequencing, total DNA was prepared using MasterPure Complete DNA Purification Kit (Epicentre; Madison, Wisconsin). NexteraXT libraries were prepared and sequenced on an Illumina NextSeq 500 at the Genomics Core at Case Western Reserve University. Paired sequence reads were mapped using bwa-mem to the DH10B genome (accession CP000948.1), the pBC SK(−) plasmid (https://www.novoprolabs.com/vector/V12548), and the SHV-1 gene (accession JX268740.1). Reads were also assembled into contigs using velvet[45]. Three approaches were used to identify de novo mutations. First, SNVs were called using the mapped reads using the Genome Analysis Toolkit (GATK)[46]. Second, large deletions were identified using a combination of detection of low-coverage regions of the reference based on read mapping results and BLAST searches between the DH10B reference sequence and the contigs. Insertion sequence (IS) elements present in the DH10B genome were identified using ISfinder[47] and locations for IS elements were mapped in the contigs using ISseeker[48].

**Reporting summary**. Further information on experimental design is available in the Nature Research Reporting Summary linked to this article.

## Data availability

All MIC measurements are available in Supplementary Dataset 1. All sequencing data are deposited to the NCBI sequence read archive under accession code PRJNA515080. The Python code used in the mathematical modelling and statistical analyses are available at: https://github.com/Daniel-Nichol/CollateralSensitivityRepeatability.

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

## Acknowledgements

D.N. would like to thank the Engineering and Physical Sciences Research Council (EPSRC) for generous funding for his doctoral studies (OUCL/DN/2013). J.G.S. is grateful to the NIH for their generous loan repayment programme and to the Paul Calabresi Career Development Award for Clinical Oncology (NIH K12CA076917). ARAA would like to acknowledge the National Cancer Institute (NCI) funded Physical Science Oncology Center grant, U54CA193489.Research reported in this publication was supported by the National Institute of Allergy and Infectious Diseases of the National Institutes of Health (NIH) under Award Numbers R01AI100560, R01AI063517, and R01AI072219 to R.A.B. This study was also supported in part by funds and/or facilities provided by the Cleveland Department of Veterans Affairs, Award Number 1I01BX001974 to R.A.B. from the Biomedical Laboratory Research & Development Service of the VA Office of Research and Development, and the Geriatric Research Education and Clinical Center VISN 10. The content is solely the responsibility of the authors and does not represent the official views of the Department of Veterans Affairs or the National Institutes of Health. The funders had no role in study design, data collection and interpretation, or the decision to submit the work for publication.

## Author contributions

D.N., R.B., and J.S. conceived of the experiment which was performed by J.R., C.B., and A.H. M.A. and S.L. performed the genomic analysis. D.N., A.A., P.J., R.B., and J.S. analysed the experimental data. D.N., P.J., A.A., and J.S. conceived of the mathematical model which was implemented by D.N. Mathematical results were analysed by D.N., P.J., A.A., and J.S. All participated in writing the manuscript.

## Additional information

**Competing interests:** The authors declare no competing interests.

