## [Peer Review File · Nature Communications]

Reviewers' comments:

Reviewer #1 (Remarks to the Author):

In this manuscript, Nichol and co-workers highlight that drug pairs that should lead to collateral sensitivity according to in vitro experiments do not always exhibit this phenomenon in practice. The reason is that resistance can arise by different mutations and mechanisms, which generally lead to different collateral sensitivity phenotypes. To make this point, the authors first use a simple model that describes evolution on a recently measured fitness landscape; the model captures the randomness inherent in this process. In this model, they quantify the likelihood of ending up in different fitness peaks and show that this can imply substantial differences in the collateral sensitivity/resistance phenotypes with other antibiotics. In a new evolution experiment performed for one drug (cefotaxime) in 12 replicates, the authors then find that different evolution replicates can indeed exhibit variable outcomes for cross-resistance or collateral sensitivity with other drugs. They conclude that the stochasticity of evolution should be taken into account by performing the relevant evolution experiments in many replicates and using the likelihood of collateral sensitivity as a measure for this effect in the future.

The general problem of identifying smart ways of using multiple drugs to minimize resistance is timely and certainly interesting for a broad audience. The present manuscript is well written and most technical aspects appear sound, in particular for the theoretical part. The potential merit of this work is that it highlights the important role of stochasticity in evolution in the context of collateral sensitivity and cross-resistance more clearly than recent papers on this topic. However, some of the results are not entirely novel (see below for more detail). Further, unlike several recent comprehensive studies in this field, the experimental aspects of this work focus on a relatively small set of drugs; this may ultimately not suffice to support some of the more general conclusions that are made. Specifically, it would be crucial to address the following issues:

1. It is well established that replicate evolution experiments often yield different results, both at the phenotypic and genotypic level. This has also been reported in previous studies of collateral sensitivity and cross-resistance, essentially whenever the relevant evolution experiments were performed with replicates (e.g. Oz, MBE, 2014 and other recent studies of this problem). While this aspect can be seen in the supplemental data of these papers, I agree with the authors that it was somewhat marginalized in these previous studies and that it is important to take this point into account. But it needs to be clarified in how far the main experimental data set of the present work (Fig. 3C) is needed and actually advances the field (the fact that these outcomes are variable is qualitatively consistent with prior observations).
2. The evolution experiment (Fig. 3) is focused on a single drug and cross-resistance / collateral sensitivity was checked with only 3 other drugs (Fig. 3C). While the number of replicates is higher, this is a limited data set compared to recent studies which usually studied complete networks of all pairwise combinations between drugs (i.e. resistance to a larger set of drugs with diverse modes of action was evolved and the sensitivity to all other drugs in the set tested; see e.g. Imamovic, Sci Transl Med, 2013; Lazar, Mol Syst Biol, 2014; Oz, MBE, 2014; Suzuki, Nat Comm, 2014). It seems that such a more comprehensive approach would also be important in the present study to support any more general conclusions.
3. There are some technical aspects of the evolution experiment that are problematic. The authors generate drug concentration gradients on agar plates and pick the most resistant colonies in each transfer. This corresponds to a small population size bottleneck, which is known to increase stochasticity in evolution. Thus, the protocol that was used may artificially over-emphasize the role of stochastic effects which may be less important in actual infections (where populations are typically large and do not go through bottlenecks). This issue is also reflected in the resistance trajectories for the different replicates (Fig. 3B), which show enormous variability (>200-fold differences in final MIC); other evolution protocols in liquid medium typically result in much lower

variability at this phenotypic level (see e.g. Toprak, Nat Gen, 2012). The protocol used by the authors also corresponds to a different regime of evolutionary dynamics than the “strong selection, weak mutation” limit that is assumed in the theoretical model (Fig. 2). Another problem with this assay is that the bacteria carry a resistance gene (beta-lactamase) on a plasmid, but mutations are not restricted to the resistance gene: While certainly more laborious, restricting mutations to the resistance gene (by re-cloning it in each transfer step) is commonly done (see e.g. Schenk, Evol Appl, 2015) and would correspond better to the theoretical model in Fig. 2. In any case, it is problematic that resistance mutations almost certainly occur also in the genome (Fig. 3B) but the authors made no attempt to identify these mutations. This would be straightforward via whole genome sequencing which is standard in the field for these types of experiments; it would corroborate the claim that the different phenotypic outcomes observed are due to different resistance mutations and may provide further insight into the mechanistic origins of these differences. For the mutations on the plasmid, a complication is that individual cells carry multiple copies of the plasmid and thus inevitably carry different versions of the mutated (or non-mutated) plasmid, leading to complicated evolutionary dynamics in which different plasmids compete with each other inside cells just as different clones compete with each other in the population. In addition, plasmid copy number increases, possibly via mutations in the replication origin, are a likely way to evolve resistance but were not checked for.

Reviewer #2 (Remarks to the Author):

MS Overview

1. Simulations are done of a theoretical landscape model showing branch points (several local fitness maxima) lead to differential outcomes from post-adaptation collateral sensitivity testing (CST).
2. Experiments are done (E.coli, 10 days, replicated) showing resistance increases to drug 1, a panel of drug 2s showing differential outcomes of collateral sensitivity (CS) testing for some drugs.
3. Authors conclude: CST can give different outcomes for some drugs, let's not lose the concept completely - but let's replicate more, be more certain about the claimed CS patterns, they need to be robust, and let's do clinical (whole?) genome sequencing to look at the cause of failures.

My Conclusion

I don't think this can be published in this format. If there was one change that might make it publishable it would if their more extensive data (in an XLS table and not shown here) showed that there were NO robust collateral sensitivities at all out of the many tested.

After all, if there's one CS there...that's the one to use. If there are none and you can show that prior claims as to their existence are false, then this would be a call to the community not to be so slap-dash in terms of when they publish claims of collateral sensitivity. After all, clinicians want to use these results (eventually) so let's not create datasets that are corrupted with noise and inconsistencies.

The lack of repeatability in empirical science is an issue of some note at the moment, after all: <https://www.nature.com/news/1-500-scientists-lift-the-lid-on-reproducibility-1.19970>

And I do agree that slap-dash-ness does apply to some papers and we've struggled to replicate what others have done....but then we had some success with the CS idea too:

<http://journals.plos.org/plosbiology/article?id=10.1371/journal.pbio.1002104>

Just a note: we tried like crazy to replicate our data from this paper with an *acr* knockout and to say that our collateral sensitivity data was contingent on the stars would be an understatement. We didn't feel that data was publishable, however. But it certainly made me aware that CS can be a very delicate feature, depending on the underlying genomics. But that one idea is not publishable in itself, I don't think.

General comments.

0. I found the writing style hard to follow, a bit like Billy Connolly where it meanders in each paragraph from one concept to the next. It makes the logic hard to follow. Things are discussed before they're mentioned (X1-X4 - these are said to be strains when they're replicates). The *E. coli* strain is not mentioned in the main text. Was it B, K12 or some other? Pathogenic or not?

0.5. The way statistics are quoted was strange: no measures of uncertainty (error bars, SD, s.e.) at any point in the text. No test given or quantitative rationale for divergent evolution in Figure 4. Just a red/pink and a blue is enough to qualify? Maybe the absence of p values throughout is deliberate?

1. Yes, I agree with the tenet of the paper. This work seems to have been stimulated by the Imamovic-Sommer STM (IS-STM) paper showing a collateral sensitivity network. I reviewed that paper too and asked them to show that one of their collateral sensitive paths through their network actually worked. They didn't do this, oddly, but the idea seemed strong enough that I didn't object as reviewer.

I don't believe their network for a second, but the core idea of how to optimise collateral sensitivity is reasonable, given the right kind of testing assay in vitro. The issue here is the latter, it's certainly not Morten Sommer's plate assay, but it's not clear how to do this robustly so that in vitro results pass over to in vivo treatments. This is always the way, after all.

So your final paragraph is right. Some of the data probably is there and not being looked at, whereas lab protocols will only get you so far with these ideas on precise / robust collateral sensitivities.

Thing is, the same can be said of MICs and drug interactions where there's only so much overlap between in vitro data and clinical outcomes. But people plough on nevertheless with in vitro testing (e.g. CLSI) because what else can you do?

You are right, though, given the importance, we all should be deriving robust outcomes from these assays.

2. The findings in the present MS are not very surprising in some sense: lab evolution of cross-resistance has between replicate variation. That a stochastic evolutionary model has this feature too is not surprising either. Is this just a population size issue that a clinical infection won't be subject to in the same way? What happens to that simulation data at say $10^{\{6,7,8,9\}}$ cells per mL in the simulations?

What we do need are robust indicators of collateral sensitivity and understanding of the molecular basis of these. e.g.

V. Lazar, G. Pal Singh, R. Spohn, I. Nagy, B. Horvath, M. Hrtyan, R. Busa-Fekete, B. Bogos, O. Mehi, B. Csorgo, G. Posfai, G. Fekete, B. Szappanos, B. Kegl, B. Papp, and C. Pal. Bacterial evolution of antibiotic hypersensitivity. *Mol Syst Biol*, 9, 10 2013.

So CS is not simply a landscape-derived phenomenon, it's something more about the physiology of the bacterium that's being carefully primed by the first drug for the second hit.

3. It's worth pointing out that bacterial genomes will accumulate novel mutations (not just SNPs of the types described here) but they will readily occur in these protocols in core metabolism, in nutrient transport and drug transport, in lipid components, redox metabolism / protection from ROS, core stress responses, we see even ribosomal changes and massive prophage polymorphisms in lab media with Ab drugs within hours (~24h+) to days.

We see these changes all the time in the lab on these timescales and whole genome sequencing will show them up.

My point is you cannot rely on targeted sequencing to explain MIC changes in Ecoli, not on these timescales. There's no reason not to see significant genomic deletions and amplifications but you've not tested for any of these.

And yet you then recommend that clinicians do so in their work! (I think, although the wording isn't clear.)

TBH I don't know why you haven't done whole-genome sequencing on some of these strains...their genomes will have changed considerably over these timescales...I think those data might well change too how you think about landscapes and "evolutionary branching" in particular in the context of bacteria. (They don't follow continuous random paths in some genome space at all IMHO, the genomic changes are too abrupt for that.)

I know of one study that does this (whole-genome):

M. M. Mwangi, S. W. Wu, Y. Zhou, K. Sieradzki, H. de Lencastre, P. Richardson, D. Bruce, E. Rubin, E. Myers, E. D. Siggia, and A. Tomasz. Tracking the in vivo evolution of multidrug resistance in staphylococcus aureus by whole-genome sequencing. *PNAS*, 104(22):9451–9456, May 2007.

Target sequencing in the clinic does give interesting insights:

J. M. A. Blair, V. N. Bavro, V. Ricci, N. Modi, P. Cacciotto, U. Kleinekathoefer, P. Ruggerone, A. V. Vargiu, A. J. Baylay, H. E. Smith, Y. Brandon, D. Galloway, and L. J. V. Piddock. Acrb drug-binding pocket substitution confers clinically relevant resistance and altered substrate specificity. *Proceedings of the National Academy of Sciences*, 112(11):3511–3516, 2015.

This shows how the MICs for a panel of drugs jumps around week by week (look at the supplementary) but I doubt most of this is due to the operon they sequenced weekly (acr).

My point is ... yes you are right that clinical sequencing gives interesting outcomes and should be done more, it's harder than you describe (eg. getting the human DNA out of the sample which could

swamp the bug DNA signal)

and you don't do very extensive sequencing here when you've got all the evolved bacteria in the freezer already.

4. I don't think saying "have more replicates" in experimental evolution experiments is a novel experimental approach. Is there something specific that can help with the process of increasing replicate number, do you think?

Reviewer #1 writes:

In this manuscript, Nichol and co-workers highlight that drug pairs that should lead to collateral sensitivity according to in vitro experiments do not always exhibit this phenomenon in practice. The reason is that resistance can arise by different mutations and mechanisms, which generally lead to different collateral sensitivity phenotypes. To make this point, the authors first use a simple model that describes evolution on a recently measured fitness landscape; the model captures the randomness inherent in this process. In this model, they quantify the likelihood of ending up in different fitness peaks and show that this can imply substantial differences in the collateral sensitivity/resistance phenotypes with other antibiotics. In a new evolution experiment performed for one drug (cefotaxime) in 12 replicates, the authors then find that different evolution replicates can indeed exhibit variable outcomes for cross-resistance or collateral sensitivity with other drugs. They conclude that the stochasticity of evolution should be taken into account by performing the relevant evolution experiments in many replicates and using the likelihood of collateral sensitivity as a measure for this effect in the future.

The general problem of identifying smart ways of using multiple drugs to minimize resistance is timely and certainly interesting for a broad audience. The present manuscript is well written and most technical aspects appear sound, in particular for the theoretical part. The potential merit of this work is that it highlights the important role of stochasticity in evolution in the context of collateral sensitivity and cross-resistance more clearly than recent papers on this topic. However, some of the results are not entirely novel (see below for more detail). Further, unlike several recent comprehensive studies in this field, the experimental aspects of this work focus on a relatively small set of drugs; this may ultimately not suffice to support some of the more general conclusions that are made. Specifically, it would be crucial to address the following issues:

1. *It is well established that replicate evolution experiments often yield different results, both at the phenotypic and genotypic level. This has also been reported in previous studies of collateral sensitivity and cross-resistance, essentially whenever the relevant evolution experiments were performed with replicates (e.g. Oz, MBE, 2014 and other recent studies of this problem). While this aspect can be seen in the supplemental data of these papers, I agree with the authors that it was somewhat marginalized in these previous studies and that it is important to take this point into account. But it needs to be clarified in how far the main experimental data set of the present work (Fig. 3C) is needed and actually advances the field (the fact that these outcomes are variable is qualitatively consistent with prior observations).*

Comment: We now make reference to previous studies that observed differences in collateral sensitivity in repeated replicates (lines: 45-59) as motivation for our study. As a result, we believe our study is now better framed as one that provides an evolutionary explanation for this phenomenon and as an exploration of the importance of addressing it. Further, to address the concern regarding the novelty of our approach, we have extended our experimental design by the addition of 48 additional replicates of experimental evolution under cefotaxime. We believe that these additional replicates better distinguish our work from previous studies, as the focus on stochasticity and quantification of error is now much clearer, rather than on the evaluation of all pairwise drug combinations (with lower replicate numbers) as in previous studies.

2. *The evolution experiment (Fig. 3) is focused on a single drug and cross-resistance / collateral sensitivity was checked with only 3 other drugs (Fig. 3C). While the number of replicates is higher, this is a limited data set compared to recent studies which usually studied complete networks of all pairwise combinations between drugs (i.e. resistance to a larger set of drugs with diverse modes of action was evolved and the sensitivity to all other drugs in the set tested; see e.g. Imamovic, Sci Transl Med, 2013; Lazar, Mol Syst Biol, 2014; Oz, MBE, 2014; Suzuki, Nat Comm, 2014). It seems that such a more comprehensive approach would also be important in the present study to support any more general conclusions.*

Comment: We agree with the reviewer’s comment that our initial experimental design was somewhat limited when compared to the previous studies highlighted. To remedy this issue, and to provide greater support for our conclusions, we have now performed an additional 48 (thus, a total of 60) replicates of evolution under cefotaxime (See new Figure 4). We feel we are now better able to demonstrate evolutionary divergence and confirm its role in driving divergent collateral response, for example, through empirical estimation of collateral sensitivity likelihoods (Figure 5(B)). Further, we have expanded the panel of second line drugs to a total of eight (Table 1). To our knowledge, this experiment now represents the largest number of parallel evolutionary replicates undertaken to date in the study of collateral response.

We would like to thank the reviewer for highlighting a number of studies we had omitted, these are now referenced and discussed in the introduction.

3. *There are some technical aspects of the evolution experiment that are problematic. The authors generate drug concentration gradients on agar plates and pick the most resistant colonies in each transfer. This corresponds to a small population size bottleneck, which is known to increase stochasticity in evolution. Thus, the protocol that was used may artificially over-emphasize the role of stochastic effects which may be less important in actual infections (where populations are typically*

large and do not go through bottlenecks). This issue is also reflected in the resistance trajectories for the different replicates (Fig. 3B), which show enormous variability (>200-fold differences in final MIC); other evolution protocols in liquid medium typically result in much lower variability at this phenotypic level (see e.g. Toprak, Nat Gen, 2012). The protocol used by the authors also corresponds to a different regime of evolutionary dynamics than the strong selection, weak mutation limit that is assumed in the theoretical model (Fig. 2).

Comment: We agree entirely with the reviewer that serial passaging using the gradient plate method induces bottlenecks of strong selection that likely increase the stochasticity of evolution, in particular by minimising heterogeneity, amplifying drift and preventing modes of population dynamics, such as clonal interference, that may be critical *in vivo*. This method of experimental evolution was chosen as it was the most commonly used in other collateral sensitivity papers, such as Imamovic and Sommer, Sci Trans. Med. (2013) and Oz, MBE (2014). Our aim was to compare like with like when evaluating the robustness of collateral sensitivities. We agree that an alternative, perhaps more clinically realistic, culture technique could yield much more stable collaterally sensitive drug pairs. We now make reference to this in the discussion (line 239) and outline that alternative culture techniques, such as the morbidostat designed by the Kishony lab, could partially remedy the issue of divergent evolution *in vitro*.

We also agree that the Strong Selection Weak Mutation model presented does not perfectly capture the evolutionary dynamics of serial passaging using the gradient plate method. We now clearly outline this caveat in our discussion (line 250) and highlight that mathematical models could help in the design of better *in vitro* evolution experiments. Despite the partial disconnect between our mathematical modelling and experimental design, we believe that our modelling serves to provide valuable intuition about the potential impact of stochasticity in evolution (that would be infeasible, requiring an astronomical number of replicates, to derive through solely experimental means).

Another problem with this assay is that the bacteria carry a resistance gene (beta-lactamase) on a plasmid, but mutations are not restricted to the resistance gene: While certainly more laborious, restricting mutations to the resistance gene (by re-cloning it in each transfer step) is commonly done (see e.g. Schenk, Evol Appl, 2015) and would correspond better to the theoretical model in Fig. 2. In any case, it is problematic that resistance mutations almost certainly occur also in the genome (Fig. 3B) but the authors made no attempt to identify these mutations. This would be straightforward via whole genome sequencing which is standard in the field for these types of experiments; it would corroborate the claim that the different phenotypic outcomes observed are due to different resistance mutations and may provide further insight into the mechanistic origins of these differences. For the mutations on the plasmid, a complication is that individual cells carry multiple copies of the plasmid and thus inevitably carry different versions of the mutated (or non-mutated) plasmid, leading to complicated evolutionary dynamics in which different plasmids compete with each other inside cells just as different clones compete with each other in the population. In addition, plasmid copy number increases, possibly via mutations in the replication origin, are a likely way to evolve resistance but were not checked for.

Comment: We have now performed whole genome sequencing for the evolutionary replicates X1-X12 (See Table 2, lines 167-186). In addition to confirming the SHV mutations identified by targeted sequencing, a number of other SNVs, insertions and deletions were identified which could have driven further divergence. However, to our knowledge, none of these mutations are associated with known mechanisms of antibiotic resistance. Further, by consideration of the depth ratios

between reads aligning to SHV and those aligning chromosomally, and between those aligning to the SHV gene and those aligning to the plasmid backbone, we are confident that SHV was not amplified in any of X1-X12 during the evolution experiment. This is now stated on line 182.

Reviewer #2 writes:

MS Overview 1. Simulations are done of a theoretical landscape model showing branch points (several local fitness maxima) lead to differential outcomes from post-adaptation collateral sensitivity testing (CST). 2. Experiments are done (E.coli, 10 days, replicated) showing resistance increases to drug 1, a panel of drug 2s showing differential outcomes of collateral sensitivity (CS) testing for some drugs. 3. Authors conclude: CST can give different outcomes for some drugs, lets not lose the concept completely - but lets replicate more, be more certain about the claimed CS patterns, they need to be robust, and lets do clinical (whole?) genome sequencing to look at the cause of failures.

My Conclusion I dont think this can be published in this format. If there was one change that might make it publishable it would if their more extensive data (in an XLS table and not shown here) showed that there were NO robust collateral sensitivities at all out of the many tested. After all, if theres one CS there... thats the one to use. If there are none and you can show that prior claims as to their existence are false, then this would be a call to the community not to be so slap-dash in terms of when they publish claims of collateral sensitivity. After all, clinicians want to use these results (eventually) so let's not create datasets that are corrupted with noise and inconsistencies.

Comment: We have now extended our experiment to 60 evolutionary replicates under cefotaxime and a total of 8 second line drugs (assayed for MIC in triplicate). We found no robust collateral sensitivities, although we do identify piperacillin as never exhibiting cross resistance across the 60 replicates. In fact, we found the likelihood of CS to be very low for all 8 second line drugs (Figure 5(B)). These additional replicates now provide clear evidence of divergent collateral response, which taken alone do not invalidate previous studies, but do suggest that previously derived datasets should be interpreted with caution and future experiments designed with stochasticity in mind.

The lack of repeatability in empirical science is an issue of some note at the moment, after all:

<https://www.nature.com/news/1-500-scientists-lift-the-lid-on-reproducibility-1.19970>

And I do agree that slap-dash-ness does apply to some papers and weve struggled to replicate what others have done. but then we had some success with the CS idea too:

<http://journals.plos.org/plosbiology/article?id=10.1371/journal.pbio.1002104>

Just a note: we tried like crazy to replicate our data from this paper with an acr knockout and to say that our collateral sensitivity data was contingent on the stars would be an understatement. We didnt feel that data was publishable, however. But it certainly made me aware that CS can be a very delicate feature, depending on the underlying genomics. But that one idea is not publishable in itself, I don't think.

Comment: Studies can be difficult to reproduce for many reasons, which poses a major difficulty for the lab attempting to undertake corroborative experiments: Are the original results incorrect or is it that we are doing something incorrect? We think that the value of our work is not simply pointing out that CS is not robust, but rather in demonstrating a biologically/evolutionarily sound

explanation for this phenomenon. Coupled with an exploration of the potential impact of evolutionary divergence (presented through modelling) and suggestions for mitigating this problem, we think our work could help others to design better experiments to explore collateral sensitivities. Finally, we note that the existence of stochasticity in evolution may seem quite obvious to those who study evolutionary theory, but it is our experience that scientists from other fields, such as microbiologists or clinicians, can have different intuition. It is for this reason that we have opted to send this manuscript to a journal with a broad, multidisciplinary readership.

We would argue that these issues are of scientific importance. Indeed, a number of evolution studies recently published in high impact journals comprise results derived from single, or few, evolutionary replicates. One such example is Imamovic (Cell, 2018) which explores phenotypic evolution of *P. aeruginosa* using single evolutionary replicates.

General comments. 0. I found the writing style hard to follow, a bit like Billy Connolly where it meanders in each paragraph from one concept to the next. It makes the logic hard to follow. Things are discussed before they're mentioned (X1-X4 - these are said to be strains when they're replicates). The E.coli strain is not mentioned in the main text. Was it B, K12 or some other? Pathogenic or not?

Comment: We have now restructured and extended the manuscript to improve the clarity. Specifically, the replicates X1-X60 are now properly introduced before they are referenced (line: 135) and the specific strain of *E. coli* is now outlined in the main text (line: 131). The references to the final replicate populations as 'strains' are now removed. Finally, we would like to refute the Billy Connolly comparison as it is demonstrably false – the manuscript contains no expletives.

0.5. The way statistics are quoted was strange: no measures of uncertainty (error bars, SD, s.e.) at any point in the text. No test given or quantitative rationale for divergent evolution in Figure 4. Just a red/pink and a blue is enough to qualify? Maybe the absence of p values throughout is deliberate?

Comment: MICs are now measured in triplicate for all replicates and all data is available as a supplementary file (Supplementary Table 1). The reported MIC values are determined via a maximum likelihood estimate following a statistical model presented by Weinreich (Science, 2006) and outlined on line 305. A likelihood ratio test is employed to determine significant changes from parental sensitivity. Collateral sensitivity and cross resistance are now reported only when the evolved line exhibits significant change in sensitivity from the parental line, following a Bonferroni correction. For full transparency, Figure 3 has been updated to show all three measurements for each replicate, the mle estimate of the MIC and an indicator of significance before/after correction for multiple hypothesis testing.

1. Yes, I agree with the tenet of the paper. This work seems to have been stimulated by the Imamovic-Sommer STM (IS-STM) paper showing a collateral sensitivity network. I reviewed that paper too and asked them to show that one of their collateral sensitive paths through their network actually worked. They didn't do this, oddly, but the idea seemed strong enough that I didn't object as reviewer.

I don't believe their network for a second, but the core idea of how to optimise collateral sensitivity is reasonable, given the right kind of testing assay in vitro. The issue here is the latter, it's

certainly not Morten Sommer's plate assay, but its not clear how to do this robustly so that *in vitro* results pass over to *in vivo* treatments. This is always the way, after all. So your final paragraph is right. Some of the data probably is there and not being looked at, whereas lab protocols will only get you so far with these ideas on precise / robust collateral sensitivities. Thing is, the same can be said of MICs and drug interactions where there's only so much overlap between *in vitro* data and clinical outcomes. But people plough on nevertheless with *in vitro* testing (e.g. CLSI) because what else can you do? You are right, though, given the importance, we all should be deriving robust outcomes from these assays.

Comment: We agree entirely with the reviewer's comments, there are myriad reasons that *in vitro* results are not robust. If we are to undertake experimental evolution with translational impact, a critical first step is to isolate each issue and attempt to remedy it. This is essentially the purpose of the submitted manuscript: to identify the lack of robustness in CS, outline one (likely of many) possible driver, quantify its impact and suggest potential remedies.

2. *The findings in the present MS are not very surprising in some sense: lab evolution of cross-resistance has between replicate variation. That a stochastic evolutionary model has this feature too is not surprising either. Is this just a population size issue that a clinical infection won't be subject to in the same way? What happens to that simulation data at say $10^{6,7,8,9}$ cells per mL in the simulations? What we do need are robust indicators of collateral sensitivity and understanding of the molecular basis of these. e.g. V. Lazar, G. Pal Singh, R. Spohn, I. Nagy, B. Horvath, M. Hrtyan, R. Busa-Fekete, B. Bogos, O. Mehi, B. Csorgo, G. Posfai, G. Fekete, B. Szappanos, B. Kegl, B. Papp, and C. Pal. Bacterial evolution of antibiotic hypersensitivity. Mol Syst Biol, 9, 10 2013. So CS is not simply a landscape-derived phenomenon, it's something more about the physiology of the bacterium that's being carefully primed by the first drug for the second hit.*

Comment: The model presented exhibits stochasticity arising from the random processes of mutation and extinction through drift. The model is applicable under the SSWM assumptions which we agree need not always hold. When these assumptions break down, for example when the population size increases, alternative dynamics can arise, such as clonal interference. What is the correct model formulation to mirror *in vivo* evolution is a key question if we are aiming for translational results, and is one that underpins much of the work in theoretical biology. As outlined above, similar issues arise in cell culture systems - what is the appropriate experimental model to derive translational results? In our discussion we now touch on how improved modelling and alternative experimental designs could derive collateral sensitivities that are generalisable to *in situ* evolutionary dynamics.

We agree entirely that drug resistance is not a solely landscape-derived phenomenon and now clearly outline this caveat in our discussion (line 257). In fact, non-genetic or physiological adaptation may play a role in our experiments, as evidenced by our new whole genome sequencing results, as we are unable to identify genomic alterations in X7, despite the significant increase in resistance (Figure 3). This observation is now discussed on lines 187 and 258.

3. *Its worth pointing out that bacterial genomes will accumulate novel mutations (not just SNPs of the types described here) but they will readily occur in these protocols in core metabolism, in nutrient transport and drug transport, in lipid components, redox metabolism / protection from ROS, core stress responses, we see even ribosomal changes and massive prophage polymorphisms in lab media with Ab drugs within hours (24h+) to days. We see these changes all the time in the lab on these timescales and whole genome sequencing will show them up. My point is you cannot rely on*

targeted sequencing to explain MIC changes in *E. coli*, not on these timescales. There's no reason not to see significant genomic deletions and amplifications but you've not tested for any of these. And yet you then recommend that clinicians do so in their work! (I think, although the wording isn't clear.) TBH I don't know why you haven't done whole-genome sequencing on some of these strains... their genomes will have changed considerably over these timescales... I think those data might well change too how you think about landscapes and "evolutionary branching" in particular in the context of bacteria. (They don't follow continuous random paths in some genome space at all IMHO, the genomic changes are too abrupt for that.) I know of one study that does this (whole-genome): M. M. Mwangi, S. W. Wu, Y. Zhou, K. Sieradzki, H. de Lencastre, P. Richardson, D. Bruce, E. Rubin, E. Myers, E. D. Siggia, and A. Tomasz. Tracking the *in vivo* evolution of multidrug resistance in *Staphylococcus aureus* by whole-genome sequencing. *PNAS*, 104(22):9451-9456, May 2007. Target sequencing in the clinic does give interesting insights: J. M. A. Blair, V. N. Bavro, V. Ricci, N. Modi, P. Cacciotto, U. Kleinekathoefer, P. Ruggerone, A. V. Vargiu, A. J. Baylay, H. E. Smith, Y. Brandon, D. Galloway, and L. J. V. Piddock. *AcrB* drug-binding pocket substitution confers clinically relevant resistance and altered substrate specificity. *Proceedings of the National Academy of Sciences*, 112(11):3511-3516, 2015. This shows how the MICs for a panel of drugs jumps around week by week (look at the supplementary) but I doubt most of this is due to the operon they sequenced weekly (*acr*). My point is... yes you are right that clinical sequencing gives interesting outcomes and should be done more, it's harder than you describe (eg. getting the human DNA out of the sample which could swamp the bug DNA signal) and you don't do very extensive sequencing here when you've got all the evolved bacteria in the freezer already.

Comment: Whole genome sequencing has now been performed for the original 12 replicates reported, as well as the parental strain. A full summary of genomic alterations identified is provided in Table 2. Although chromosomal SNVs, deletions and insertions were identified, none were in genes known to confer antibiotic resistance. Further, no amplification of SHV was evident. The identification of additional genomic alterations stratifying our replicates lends further credence to our hypothesis: that evolution is a stochastic and not necessarily repeatable process. We agree that the fitness landscape metaphor is imperfect as it fails to account for large-scale genomic events. However, it is not our intention to fully capture biological reality with our modelling, but rather to provide intuition regarding the extent to which stochasticity in evolution is a confounding factor in the present methodology of collateral sensitivity studies.

4. *I don't think saying have more replicates in experimental evolution experiments is a novel experimental approach. Is there something specific that can help with the process of increasing replicate number, do you think?"*

Comment: We agree that "have more replicates" is not, alone, particularly insightful. Our work has now been extended to explicitly follow our own advice, and we now derive empirical collateral sensitivity likelihoods from 60 replicates (Figure 4). Further, drawing from the suggestions of the reviewers, we have extended the discussion to suggest how better to minimise stochasticity and increase replicate numbers in experimental evolution (line 232 onwards). We suggest alternative, automated culture techniques (to remedy both the bottle-neck induced stochasticity and the difficulty in scaling up replicate number), identify the potential for culture conditions to more closely mirror *in situ* dynamics (no bottlenecks, correct drug concentrations, etc) and discuss the potential for improved mathematical models (line 250) to provide intuition and insight to aid in experimental design.

We would like to conclude by again thanking the editors and reviewers for their time and effort.

REVIEWERS' COMMENTS:

Reviewer #2 (Remarks to the Author):

Review response for both reviewers.

First, I must express my apologies for the array of flippant comments in the first review. I blame my 12 month old daughter and lack of sleep at the time of writing.

Nevertheless, the comments between reviewers were relatively consistent as to aspects of the article that needed addressing, which were these (to approximation):

Note to editor: CS means "antibiotic collateral sensitivity" throughout the text below.

1. The article was IMHO slightly hyperbolic, in parts, in attacking the concept of CS by using applying an in vitro experimental protocol, although one often used by others, therefore some justification of the rationale for the chosen protocol seemed necessary.
2. Indeed, the nature of this protocol induces a high degree of variability / drift through bottlenecking and this was a main feature of the claims overall: a lack of repeatability in CS data/patterns.
3. No genomic analysis had been done, despite saying others should do it (!), and reviewers asked if plasmid / chromosomal copy number variants /SNPs could induce off-target resistance.
4. As the landscape model only accounted for the Ab target, it could mislead by not accounting for those additional (off target) mutational affects.

—

To address these in turn:

1. Yes, the new additions to the text in red do tone down the claims and clarify the article.

The authors also respond that their choice here is motivated by the papers (Imamovic and Sommer, Sci Trans. Med. (2013) and Oz, MBE (2014)) and I certainly agree - we could not replicate the data in the first of these in our lab and you are highlighting one reason for that. But (seeking some generality) we used a different protocol, namely the one in

<http://journals.plos.org/plosbiology/article?id=10.1371/journal.pbio.1002104>

Given that reference, and others, the word "Frequently" seems slightly misplaced on p3, l37.

I'd just like a small sentence added here to reflect the fact that you're making a choice here between different prior in vitro protocols and so you plump for the Imamovic and Sommer one - with some rationale.

You do mention other protocols later on, like the morbidostat, but the choice of lab protocol is clearly important here, given the strong bottlenecking. Like all systems, your findings could be system specific. Ok, the model works to mitigate that, but I'd like clarity around the idea that there are different CS protocols and, if this idea of CS is to be clinically relevant, we need to discern which of

these lab possibilities is likely to provide clinically useful information.

2. Yes, there is bottlenecking but you explain this as borrowing what others have done.

3. You've now undertaken some genomics. It is interesting that you do see deletions, though you claim no relevant genomic or plasmid copy number variation for Ab resistance.

But - I notice an OmpC mutation in your table, and although the function of the mutated base pair escapes me, OmpC facilitates beta lactam uptake:

<https://www.ncbi.nlm.nih.gov/pmc/articles/PMC185697/>

Membrane-altering mutations are likely to interact with beta lactams and given your plate protocol,

I'd be interested to know if those deletions also arose in a control protocol implemented without antibiotic,

but that work hasn't been done...if indeed it makes sense to do this in the absence of antibiotic.

One for another day...

My guess (and it is a guess) is the deletions may regulate resistance, for example through membrane

structure, but you are right in claiming that the impact on resistance of these mutations is not known.

Returning to a topic that Billy Connolly may have some interest in, Tom Ferenci calls all this SPANC

balance - the idea that there is a tradeoff between Stress Protection And Nutritional Competence - whereby reduced permeability to small molecules promotes resistance but reduces nutrient uptake.

This is a mechanistic source of the costs of resistance that you allude to, although fitness cost data for AbR seems to me to be much less clear cut than you state it to be. I note that you don't cite any papers there but the totality of data in the literature on costs seems a little confusing to me, I still don't know if they exist, or not. I think it's more a case that you can find them, but you can also find "cost-busting" mutations on occasion i.e. dual benefits. So costs can go either way, from what I see in the lit. (And then there's Dan Andersson's work on downstream compensation of fitness costs due to genomic amplifications that's not cited on that point either.)

4. The changes in language have mollified claims around the model and it's good to discuss limitations, so that seems fine.

note

p14 I239 seems incomplete ... "to track and manipulate" what?

—

Conclusion

I.

Given all this, I'd conclude that "generalisability" of the in vitro findings are important relative to the claims so I'd still like that point 1 above addressed at the start whereby that word "Frequently" (p3, I37)

could be made a little more precise. I'd like to see something stated along the lines that different

protocols are used to assess CS in the literature and that this paper deals with one of the first of these implemented to determine CS patterns, namely....the one Imamovic and Sommer use.

It seems in keeping with the thrust here that if CS is to be taken seriously in infection medicine, we need some standardisation of lab protocols (as was done for MICs) if we are to best discern where the robust CS patterns are.

(As a side-note, the standardisation and quantification of data around 2-drug Ab interactions was (and still is) something of a struggle in the literature and CS has the strong potential to become a similarly confusing topic if we're not careful.)

II.

It is worth stating somewhere briefly whether that particular mutation in OmpC is known to have an effect (or else is known to have none) on antibiotic uptake or resistance. Otherwise, a general statement about omp's and Ab resistance seems to be needed somewhere. Maybe even SPANC, in that case.

Indeed, given the function of envZ in regulating outer membrane protein expression (<https://www.ncbi.nlm.nih.gov/pubmed/11973328>) it seems relevant too.

We thank the reviewer for their comments and suggestions and have made additional changes to the manuscript in response. These changes are highlighted blue in the manuscript in order to distinguish them from changes made in response to the suggestions of the editor. Below we first copy the comments of the reviewer followed by a list of changes we have made to accommodate these comments.

The reviewer writes:

First, I must express my apologies for the array of flippant comments in the first review. I blame my 12 month old daughter and lack of sleep at the time of writing. Nevertheless, the comments between reviewers were relatively consistent as to aspects of the article that needed addressing, which were these (to approximation):

Note to editor: CS means "antibiotic collateral sensitivity" throughout the text below.

1. The article was IMHO slightly hyperbolic, in parts, in attacking the concept of CS by using applying an in vitro experimental protocol, although one often used by others, therefore some justification of the rationale for the chosen protocol seemed necessary.
2. Indeed, the nature of this protocol induces a high degree of variability / drift through bottlenecks and this was a main feature of the claims overall: a lack of repeatability in CS data/patterns.
3. No genomic analysis had been done, despite saying others should do it (!), and reviewers asked if plasmid / chromosomal copy number variants /SNPs could induce offtarget resistance.
4. As the landscape model only accounted for the Ab target, it could mislead by not accounting for those additional (off target) mutational effects.

To address these in turn:

1. Yes, the new additions to the text in red do tone down the claims and clarify the article. The authors also respond that their choice here is motivated by the papers (Imamovic and Sommer, *Sci Trans. Med.* (2013) and Oz, *MBE* (2014)) and I certainly agree we could not replicate the data in the first of these in our lab and you are highlighting one reason for that. But (seeking some generality) we used a different protocol, namely the one in

<http://journals.plos.org/plosbiology/article?id=10.1371/journal.pbio.1002104>

Given that reference, and others, the word Frequently seems slightly misplaced on p3, l37. I'd just like a small sentence added here to reflect the fact that you're making a choice here between different prior in vitro protocols and so you plump for the Imamovic and Sommer one with some rationale. You do mention other protocols later on, like the morbidostat, but the choice of lab protocol is clearly important here, given the strong bottlenecks. Like all systems, your findings could be system specific. Ok, the model works to mitigate that, but I'd like clarity around the idea

that there are different CS protocols and, if this idea of CS is to be clinically relevant, we need to discern which of these lab possibilities is likely to provide clinically useful information.

2. Yes, there is bottlenecking but you explain this as borrowing what others have done.
3. Youve now undertaken some genomics. It is interesting that you do see deletions, though you claim no relevant genomic or plasmid copy number variation for Ab resistance.

But I notice an OmpC mutation in your table, and although the function of the mutated base pair escapes me, OmpC facilitates beta lactam uptake:

<https://www.ncbi.nlm.nih.gov/pmc/articles/PMC185697/>

Membrane altering mutations are likely to interact with beta lactams and given your plate protocol, Id be interested to know if those deletions also arose in a control protocol implemented without antibiotic, but that work hasnt been done...if indeed it makes sense to do this in the absence of antibiotic. One for another day...

My guess (and it is a guess) is the deletions may regulate resistance, for example through membrane structure, but you are right in claiming that the impact on resistance of these mutations is not known. Returning to a topic that Billy Connolly may have some interest in, Tom Ferenci calls all this SPANC balance the idea that there is a tradeoff between Stress Protection And Nutritional Competence whereby reduced permeability to small molecules promotes resistance but reduces nutrient uptake. This is a mechanistic source of the costs of resistance that you allude to, although fitness cost data for AbR seems to me to be much less clear cut than you state it to be. I note that you dont cite any papers there but the totality of data in the literature on costs seems a little confusing to me, I still dont know if they exist, or not. I think it's more a case that you can find them, but you can also find "cost busting" mutations on occasion i.e. dual benefits. So costs can go either way, from what I see in the lit. (And then there's Dan Andersson's work on downstream compensation of fitness costs due to genomic amplifications that's not cited on that point either.)

4. The changes in language have mollified claims around the model and its good to discuss limitations, so that seems fine.

note p14 l239 seems incomplete ... to track and manipulate what?

Conclusion

I. Given all this, Id conclude that generalisability of the in vitro findings are important relative to the claims so Id still like that point 1 above addressed at the start whereby that word Frequently (p3, l37) could be made a little more precise. Id like to see something stated along the lines that different protocols are used to assess CS in the literature and that this paper deals with one of the first of these implemented to determine CS patterns, namely....the one Imamovic and Sommer use.

It seems in keeping with the thrust here that if CS is to be taken seriously in infection medicine, we need some standardisation of lab protocols (as was done for MICs) if we are to best discern where the robust CS patterns are.

(As a sidenote, the standardisation and quantification of data around 2drug Ab interactions was (and still is) something of a struggle in the literature and CS has the strong potential to become a similarly confusing topic if were not careful.)

II. It is worth stating somewhere briefly whether that particular mutation in OmpC is known to have an effect (or else is known to have none) on antibiotic uptake or resistance. Otherwise, a general statement about omps and Ab resistance seems to be needed somewhere. Maybe even SPANC, in that case. Indeed, given the function of *envZ* in regulating outer membrane protein expression (<https://www.ncbi.nlm.nih.gov/pubmed/11973328>) it seems relevant too.

Changes summary:

1. The word ‘Frequently’ on page 3 has been replaced by ‘One common protocol’ to emphasise that this methodology is used widely but is by no means the only protocol in common use.
2. p14, line 237 has been updated to read ‘track and manipulate evolution’.
3. We now outline in the discussions (line 253) that our results are restricted only to the gradient plate method, that other protocols exist (i.e. Fuentes-Hernandez et al. 2015) and that experimental evolution in such a system may be more appropriate for collateral sensitivity analysis.
4. We now discuss the function of *envZ* and *ompC* and highlight that mutations could drive resistance through altered drug uptake, citing the appropriate literature (line 174). We also discuss the concept of the SPANC trade-off, which could imply a cost in a drug-free environment. Interestingly this kind trade-off is being exploited in cancer therapy through adaptive therapies. This suggests that evolutionary theorists working in both fields (and others) could gain from working more closely with one another.